# Confidence reports during perceptual decision making dissociate from changes in subjective experience

Nicolás Sánchez-Fuenzalida [1,2,3,4] ✉, Simon van Gaal[1,2], Stephen M. Fleming [5,6], Julia M. Haaf [7] & Johannes J. Fahrenfort [1,2,3,4]

In noisy perceptual environments, people frequently make decisions based on non-perceptual information to maximize rewards. Therefore, a central problem in psychophysics, metacognition and consciousness research is to distinguish between decisions resulting from changes in subjective experience and those arising from non-perceptual information. It has recently been proposed that confidence reports can be used to discriminate between changes in subjective experience and those arising from non-perceptual information. Here we use a Bayesian ordinal modelling framework combined with an explicit measure of subjective experience to show across two experiments ($N = 204$) and three bias manipulations that confidence during perceptual decision-making does not uniquely reflect subjective experience. Instead, non-perceptual manipulations affecting response bias 'leak' into perceptual confidence reports. This occurs not only for biases resulting from changes in the base rate of stimuli ('cognitive' priors), but also when biasing information does not inform decision correctness (asymmetric payoff matrix). The relative strength of biases in first-order responses and confidence may help disentangle whether a given bias manipulation is perceptual in nature or not.

Humans combine both perceptual and non-perceptual information when making perceptual decisions. For example, when facing uncertain sensory information, an airport security guard may be inclined to report certain shapes on an X-ray as a potential gun, given the costly consequences of allowing such an object to go into a plane, even if the perceptual evidence is meagre. Alternatively, you may be pondering whether the person sitting next to you in the train is that famous actor from TV, but decide against it because the odds of such an event are slim, despite the perceptual evidence being ample. What these two examples have in common is that it is not immediately clear how to assess whether these decisions stem from changes in observers' subjective experience, or from a non-perceptual decisional process. Indeed, changes in categorisation behaviour may occur even if observers' perceptual experience remains the same[1]. A central challenge in research on perceptual decision-making is therefore to separate decisions reflecting the subjective experience of observers from decisions reflecting response bias. Traditionally, this problem has been addressed through the development of signal detection theoretic (SDT) models that dissociate sensitivity (d') from bias (criterion) in decision making[2]. Although these

parameters were not intended to have psychological meaning, previous research has implicitly equated d' effects with perceptual changes and criterion effects with non-perceptual changes. However, the criterion parameter in these models can reflect both perceptual or non-perceptual changes in response behaviour (see ref. 3 for a detailed account). We have recently tackled this problem by asking observers to reproduce their experience in a controlled fashion, thereby distinguishing between experimental manipulations that affect perceptual experience (visual illusions) from those that only affect response bias (base rate and payoff manipulations)[4].

There is also a long tradition of using confidence judgments to assess changes in subjective experience. Aside from its use to build Receiver Operating Characteristic curves in SDT, confidence has come to be frequently used within consciousness science as a marker of the strength of perceptual experience[5–8]. The idea that observers have privileged access to their own experience, and that this experience can be gauged through confidence judgements, is intuitively appealing. Put simply, decisions based on strong sensory evidence are more likely to be correct and elicit a greater sense of confidence, while decisions based on weak sensory evidence are

[1]Department of Psychology, University of Amsterdam, Amsterdam, The Netherlands. [2]Amsterdam Brain & Cognition, University of Amsterdam, Amsterdam, The Netherlands. [3]Department of Applied and Experimental Psychology, Free University Amsterdam, Amsterdam, The Netherlands. [4]Institute for Brain and Behavior Amsterdam, Free Universities Amsterdam, Amsterdam, The Netherlands. [5]Department of Experimental Psychology, University College London, London, United Kingdom. [6]Max Planck Centre for Computational Psychiatry and Ageing Research, University College London, London, United Kingdom. [7]Department of Psychology, University of Potsdam, Potsdam, Germany. ✉e-mail: n.a.carvajalsanchez@vu.nl

likely to be incorrect and elicit greater uncertainty. Indeed, a large body of research has underscored the ability of observers to accurately track their performance on perceptual discrimination tasks using confidence ratings–known as metacognitive sensitivity[9–14]. Accordingly, it has recently been proposed that confidence reports can be used as a diagnostic tool to discern between perceptual and non-perceptual effects on decision-making[15].

However, while it is true that both first-order judgements (decisions about external stimuli) and second-order judgements (confidence ratings in one's own performance) heavily rely on sensory input, both can potentially be affected by non-perceptual information[6,16–19]. For example, if one knows that an event is more likely to occur, one can report the prevalent event while also reporting high confidence, even if the true identity of the stimulus is completely unknown. While this behaviour can be considered normatively adaptive (as the most prevalent option is more likely to be correct), it also highlights how non-perceptual information can affect confidence judgements. Further, there is also evidence to suggest that changes in the reward value associated with a decision can bias confidence reports[20,21], even when the reward does not provide a relevant cue to decision accuracy. However, none of these experiments have used an independent benchmark to establish whether shifts in confidence coincide with changes in subjective experience, which is necessary to determine whether any such shifts are perceptual or non-perceptual in nature. For example, biases in confidence due to asymmetrical reward may be interpreted as non-perceptual, but they might also reflect a change in perceptual experience, despite the manipulation appearing to be non-perceptual. Without an explicit benchmark measure of subjective experience, this is impossible to determine. In light of this range of possible non-perceptual contributions to confidence and the unknown relationship between confidence and subjective experience, it remains undetermined how and under what circumstances subjective experience is faithfully tracked by confidence reports.

To address this question, we used a perceptual decision-making task in combination with three different bias manipulations. Critically, we also included a bias-independent measure of subjective experience to establish the relationship between confidence and experience. To this end, we recently validated a measure ('controlled reproduction') that can be used to establish whether a given manipulation affects subjective experience, independently from effects of response bias[4]. Controlled reproduction is a variant of a broader category of matching tasks that allow one to measure how a stimulus appears to an observer (appearance-based procedures, see Kingdom et al.[22]). Here, we use this measure as a benchmark to determine whether biases in confidence are diagnostic of changes in subjective experience or not. Across two experiments, we employed three manipulations, each reflecting a different bias source: a perceptual manipulation (Müller–Lyer illusion), a punishment scheme (asymmetric payoff matrix), and changes in the ratio of relevant stimuli (base rate manipulation; see Fig. 1A for a graphical depiction of the manipulations). If confidence reports uniquely track subjective experience, we expect perceptual manipulations (Müller–Lyer) to influence both first-order decisions and confidence reports, whereas non-perceptual manipulations (base rate and payoff) should only influence first-order decisions[4]. If, on the other hand, confidence reports capture both changes in subjective experience and non-perceptual decision biases, we expect all three manipulations to influence first-order decisions as well as confidence reports.

## Methods

Method and analysis are the same for experiments 1 and 2 unless stated otherwise. The experiments were not preregistered.

### Ethics

All experimental procedures were approved by the University of Amsterdam Ethics Review Board (ref. number 2021-BC-14131). Informed consent was obtained in accordance with the ethics board-approved procedures.

### Participants

122 participants (93 women and 29 men, 21.3 years old on average, SD = 3.3) and 123 participants (104 women and 19 men, 20.5 years old on average, SD = 2.8) took part in experiments 1 and 2, respectively. In both experiments, gender information was reported by participants. All participants were recruited through the University of Amsterdam lab pool and were guaranteed to be rewarded 1.5 research credits (or 15 euros). On top of this, participants could earn an extra reward (up to 0.5 research credits or 5 euros) that depended on their performance during the experiment. On average, the total payment was similar across conditions. Each participant completed one session of roughly 90 minutes in the Müller–Lyer and payoff condition, or 120 minutes in the base rate condition, including instructions, practice, and breaks.

Since all the confirmatory analyses in this experiment were conducted in a Bayesian framework, in experiment 1, we collected the data of 30 participants on each of the three bias manipulation conditions, removed outliers, removed participants with poor psychometric curve fits and then ran a Bayesian $t$-test between the biased to long and short conditions in the decision, confidence and in the reproduction task (see below for details on the removal of outliers and participants with poor fits). If there was moderate evidence for the effect of our manipulation in all tasks (for either the null or the alternative hypothesis), we stopped data collection ($BF_{10} > 3$ or $BF_{10} < 0.3$), otherwise we collected five more subjects and repeated the process. In experiment 2 we aimed to collect the same number of participants as in experiment 1, given that both experiments were identical aside from the moment at which participants provided their confidence rating. In this framework, optional stopping or data peaking is not considered problematic[23].

Participants were considered outliers if their signal detection theory criterion, d' or reproduction error, fell outside four standard deviations from the sample mean, that is, around the grand average across bias manipulation conditions. Participants with a signal detection theory d' below zero were also filtered out. In experiment 1 two participants were removed (one reproduction error outlier in the base rate condition and one participant with d' < 0 in the Müller–Lyer condition). In experiment 2 four participants were removed (one reproduction error outlier in the base rate condition and three participants with d' < 0, two in the payoff condition and one in the Müller–Lyer condition). Participants were also excluded based on curve fit quality (see Analysis–Curve fitting). In experiment 1, eight participants were removed in the Müller–Lyer condition and four in the base rate condition. In experiment 2, ten participants were removed in the Müller–Lyer condition, eleven in the base rate condition, and three in the payoff condition.

### Tasks

On each trial, participants had to either categorise a line as being shorter or longer than a reference line and report their confidence on this decision (decision/confidence task) or they had to reproduce the length of the lines presented to them (controlled reproduction task). Crucially, observers did not know which task they would be performing while they viewed the target line, thus preventing specific task demands from affecting stimulus processing. Confidence was either provided concurrently with their categorisation decision (experiment 1) or after the categorisation decision was submitted (experiment 2). In experiment 1, concurrent decision-confidence judgements were provided using a single response for both confidence and decision. The left hand was used for short decisions: the middle finger for high-confidence short, the index finger for low-confidence short. The right hand was used for long decisions: index finger for low confidence long, and middle finger for high confidence long (see Supplementary Fig. S1 for the exact key mappings in experiment 1). In experiment 2, both decision and confidence were entered using the left hand. The decision for short was given using the ring finger (left key), and the decision for long was given using the index finger (right key). After the decision, a confidence report was given on orthogonally oriented keys, using only the middle finger for either high confidence (upper key) or low confidence (lower key)

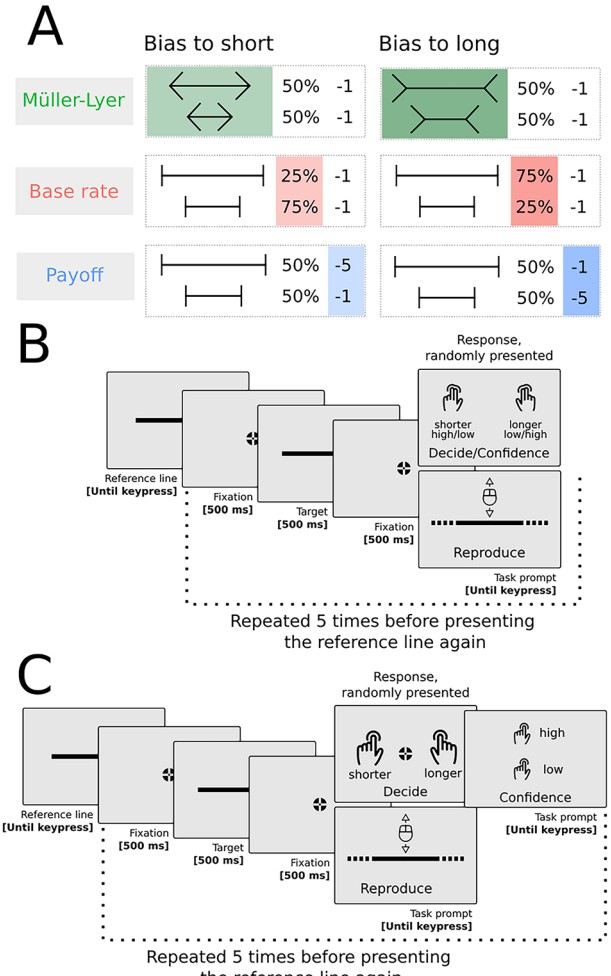

**Fig. 1 | Trial layout and bias manipulations summary. A** Manipulation summary: Target lines presented in the Müller–Lyer condition were flanked by inward-pointing arrowheads when the bias direction was long or by outward-pointing arrowheads when the bias direction was short. In the base rate and payoff condition, vertical lines flanked the target lines. In the base rate condition, there were three times more long lines than short lines when the bias direction was long, and vice versa when the bias direction was short. In the Müller–Lyer and payoff conditions, there were an equal number of long and short trials. In the payoff condition, participants lost 5 points for incorrectly answering long and 1 point for incorrectly answering short when the bias direction was short, and vice versa when the bias direction was long. The green-red-blue colour scheme is used consistently across figures to denote the different bias manipulations. **B** Experiment 1 trial layout: A typical sequence of trials, from here onward referred to as a mini-block, consisted of the presentation of a reference screen (until key press) followed by five trials. Each trial consisted of a fixation period (500 ms), followed by a target screen (500 ms), followed by a second fixation period (500 ms). At the end of the trials observers were prompted to either categorise the target line they just saw as being shorter or longer than the reference line (short/long), and to concurrently provide their confidence on their response (high/low), or to reproduce the length of the target line by increasing/decreasing a line by moving the mouse (the prompt was shown until an answer was registered). **C** Experiment 2 trial layout: Same as A), but observers reported their confidence in their response after the categorisation response. The icons depicting a two-finger ("Top Finger Tap") and one-finger ("Tap") key-press, and the icon depicting a computer mouse ("Mouse") were made by Gregor Cresnar from the-nounproject.com (licensed under CC BY 3.0).

(see Supplementary Fig. S2 for key mappings in experiment 2). See Fig. 1B, C for the trial layout of each experiment. The instructions participants received regarding how to report confidence were similar to those in the experiment by Gallagher et al.[15]. (for the full decision-confidence instructions see Supplementary Figs. S1 and S2).

## Stimuli

Stimuli consisted of black horizontal lines presented at the centre of the screen over a grey background. The reference line was 400 pixels long, and there were seven possible target lengths that ranged between 370 and 430 in steps of 10 pixels. All stimuli were presented on a 23" (58.4 cm) monitor with a resolution of 1920 × 1080 pixels, at a distance of ~75 cm. At this distance, the size of each pixel was 0.265 mm, or 0.02 visual angle degrees. The monitor refresh rate was 120 Hz. In length reproduction trials, the reproduction line was initially presented with a length of 40 pixels, bounded between 0 and 800 pixels. In the Müller–Lyer condition, each diagonal line that formed the arrowheads on the target line was 60 pixels long and subtended a 45- or 135-degree angle with the horizontal line. The vertical flanking lines in the payoff and base rate conditions were 70 pixels long. All lines had a width of 4 pixels. On each trial, a fixation period (500 ms) was followed by the target line (500 ms), further followed by a second fixation period (500 ms). The experiment was programmed on Python 3.6[24] and PsychoPy (v2020.2.10[25]).

## Design

We employed a 3 between-participant bias source (Müller–Lyer illusion, payoff, and base rate) x 2 nested within-participants bias direction (biased to long and biased to short) design. The three bias conditions were identical except for the following details: (1) In the Müller–Lyer condition, target lines were flanked by outward-pointing arrowheads when the bias direction was short, or inward-pointing arrowheads when the bias direction was long. In the payoff and base rate conditions, target lines were flanked by vertical lines. (2) In the base rate condition, the ratio of target lines that were longer or shorter than the reference line was uneven, so one category was three times more likely to be presented (see Supplementary Fig. S3 for a histogram describing the frequency of each target length value). In the payoff and Müller–Lyer condition, the ratio between target lines that were longer or shorter than the reference was even. (3) In the payoff condition, participants were differentially punished for incorrect responses in the decision task. If the bias direction was short, incorrect 'long' responses cost them five times more than incorrect 'short' responses, and vice versa when the bias direction was long. In the base rate and Müller–Lyer condition, there was a flat penalty of one point for any incorrect response in the decision task. See Fig. 1A for a graphical depiction of these bias manipulations. For all conditions, mistakes in the reproduction task that were too far off were always punished with one point, regardless of the direction of the error (see below for details).

## General procedures

For each task, participants received extensive instructions and practice (see Supplementary text T1 for a detailed description). During the experiment, the reference line was presented every five trials (see Fig. 1B, C). In the base rate condition, the relative frequency of short and long lines was presented next to the reference line, while in the payoff condition, the cost for incorrectly answering 'short' and 'long' was presented next to the reference line. In the Müller–Lyer condition, no extra information was presented with the reference line (see Supplementary Fig. S4 for an example of the reference screen and target screen of each condition). The presentation of the categorisation or reproduction prompt was counterbalanced. After every 50 trials, participants would receive block-level feedback about their performance in the decision and reproduction task during the previous 50 trials. The block-level feedback they received about the decision task was different for the different conditions. In the payoff and base rate condition, they would be shown how often they had incorrectly answered 'short' or 'long'. In the Müller–Lyer condition they were informed about the overall number of mistakes in the previous 50 trials. The block-level feedback regarding their performance in the reproduction task was the same for all conditions and always consisted of the overall number of reproduction errors. A reproduction 'error' was defined as a deviation of more than 22 pixels from the actual length of the target line, regardless of the direction of the error (over or underestimation), and it was merely provided to motivate participants to reproduce as accurately as they could. The 22 pixel deviation was based on

the data from our previous study[4] as this threshold would yield on average above chance performance. Additionally, participants received feedback on their relative use of low and high confidence reports. Participants received an extra message reminding them to use the low and high confidence option approximately evenly if they answered 'low' or 'high' in less than 25% of the trials of a 50-trial block. An example of a block-level feedback screen is provided in Supplementary Fig. S5.

The experiment was divided into two bias direction blocks (the order was counterbalanced so roughly half of the participants started with the biased to long condition). In the Müller–Lyer and payoff condition observers completed 280 decision trials and 140 reproduction trials per bias direction, summing up to 840 in total (420 trials per bias direction). Each of the seven target lines was presented 40 times in the decision/confidence task and 20 in the reproduction task. In the base rate condition observers completed 1170 trials, where the decision/confidence and reproduction task was respectively presented 390 and 195 times. In the decision/confidence task, depending on the bias direction, the targets in the more prevalent category were presented 120, 90 and 60 times going from the farthest (longest/shortest) to the closest length to the reference line (see Supplementary Fig. S3 for a histogram describing the frequency of each target length value). To increase the effect of the base rate manipulation we made the shortest or longest target lines more prevalent within the prevalent category (short/long). The target lines of the non-frequent category and the target that had the same length of the reference line were presented 30 times each. In the reproduction task each target was presented half of the times as described for the decision/confidence task. Every 50 trials participants received block-level feedback and could choose to have a break or to continue immediately. In addition, they were forced to take a 3 min break every 200 trials.

## Analysis

**Signal detection analysis**. To determine performance and bias on the tasks we computed signal detection sensitivity ($d'$) and criterion ($c$) based on hit rate and false alarms as follows:

$$d' = Z(HR) - Z(FAR)$$

and

$$c = 1/2 \times (Z(HR) + Z(FAR))$$

Where $Z()$ denotes the inverse of the standard normal cumulative distribution (often denoted as the Z-transform, as it has a mean of 0 and a standard deviation of 1). The formula can be easily translated to R code by replacing the $Z()$ with the *qnorm()* function from the R stats package. HR denotes hit rate, FAR denotes false alarm rate. In this setting correct 'long' responses are considered hits and correct 'short' responses correct rejections.

**Curve fitting**. For the decision data (categorisation task) we fitted a Cumulative Gaussian function to each participant's distribution of 'short' and 'long' responses as a function of the length of the target lines separately for each bias source and bias direction condition. The point of subjective equality (PSE), that is, where the probability of answering 'short' was 50%, was interpolated from each fitted curve. For the confidence data we fitted a quadratic function to each participant's distribution of 'low' and 'high' responses as a function of the length of the target lines separately for each bias source and bias direction condition. The point of peak uncertainty was then interpolated by identifying the lowest confidence point for each fitted curve. For the reproduction data, we fitted a linear function to each participant's distribution of length reproductions as a function of the length of the target lines separately for each bias source and bias direction condition. We then interpolated the point at which reproduction responses matched the length of the reference line (400 pixels). Participants were excluded from the main analysis

if their logistic coefficient in the decision task, quadratic coefficient in the confidence task or linear coefficient in the reproduction task was not significantly different than zero or if it was negative.

**Bayesian model comparison**. We adopted the Bayesian model comparison framework to test for ordinal-constrained models. This framework allows one to turn relations that are articulated verbally into models of ordinal relations (e.g., condition A > B, etc.). These statistical models can then be compared using Bayes factor model comparison (see ref. [26] for an introduction). The ordinal-constraint approach is described in Haaf et al.[27], and is based on Klugkist et al.[28], encompassing prior approach. We started with an unconstrained model (model A) that consisted of all three manipulations having an effect (bias-to-long > bias-to-short), and from there, we devised alternative models where one or more conditions did not have an effect (models B through G). For the unconstrained model, we use a g-prior approach as described in Rouder et al.[29], with a default setting on the scale of effect, $r = 0.707$. The other models are restricted versions of the unconstrained model using ordinal and equality constraints. For the analysis, we used the BayesFactor package in R[30]. For a graphical depiction of the models, see Supplementary Fig. S6.

**Metacognitive efficiency**. For the decision data (categorisation task) we estimated metacognitive efficiency by estimating the log(meta-$d'$/$d'$), or M-ratio, where $d'$ corresponds to the first-order sensitivity, as estimated using standard signal detection theory, and meta-$d'$ corresponds to the second-order sensitivity, or the degree to which an observer discriminates correct from incorrect responses[31]. Meta-d' is in the same scale as d', but rather than reflecting type 1 sensitivity, it provides insight into how much information from hypothetical type-1 judgements the observer was able to convert into accurate confidence judgements. Although meta-d' gives insight into the metacognitive sensitivity of an observer, it is confounded by type 1 performance, as it increases or decreases as type 1 sensitivity goes up or down. This can be ameliorated by computing M-ratio, which is the ratio between meta-d' and d' as a measure of the metacognitive efficiency of the observer, reflecting the metacognitive performance of the observer irrespective of their type-1 sensitivity. To estimate M-ratio we used the HMeta-d package[32]. For each condition (bias source by bias direction combination), four Markov chain Monte Carlo samplers of 20,000 iterations each and a thinning interval of 2 were used. The R-hat values were smaller than 1.02 for both group- and subject-level estimations, indicating satisfactory convergence of the chains. Trials where the target line had the same length as the reference line were not included in this analysis as they could not be coded as belonging to the long lines nor to the short lines class.

**Savage-Dickey density ratio method**. To determine whether there was a difference in metacognitive efficiency across bias-to-long and bias-to-short conditions we used the Savage-Dickey density ratio method[33] (see also ref. [34] for a detailed account in the context of psychological research). By using this approach, it is possible to obtained a Bayes Factor by dividing the height of a posterior distribution for a given value of interest θ by the height of the prior for θ. In our case, we were interested in knowing the likelihood of the difference in M-ratio between bias-to-long and bias-to-short being zero. Therefore, we sampled a posterior distribution of M-ratio values for the bias-to-long and bias-to-short conditions and use it to generate a posterior distribution of effect sizes (Cohen's d). For the prior we used a Cauchy distribution with a scale of 0.707 (as in all our previous analysis). Then, to assess whether the null or the alternative hypothesis was more likely, we simply divided the height of the posterior distribution of effect sizes for the value zero by the height of the prior for the value zero. Bayes Factor obtained this way can be read in the same way as all other Bayes Factors reported in this study, i.e. $BF_{10} = 3$ indicates that the data is 3 times more likely under the alternative hypothesis than under the null.

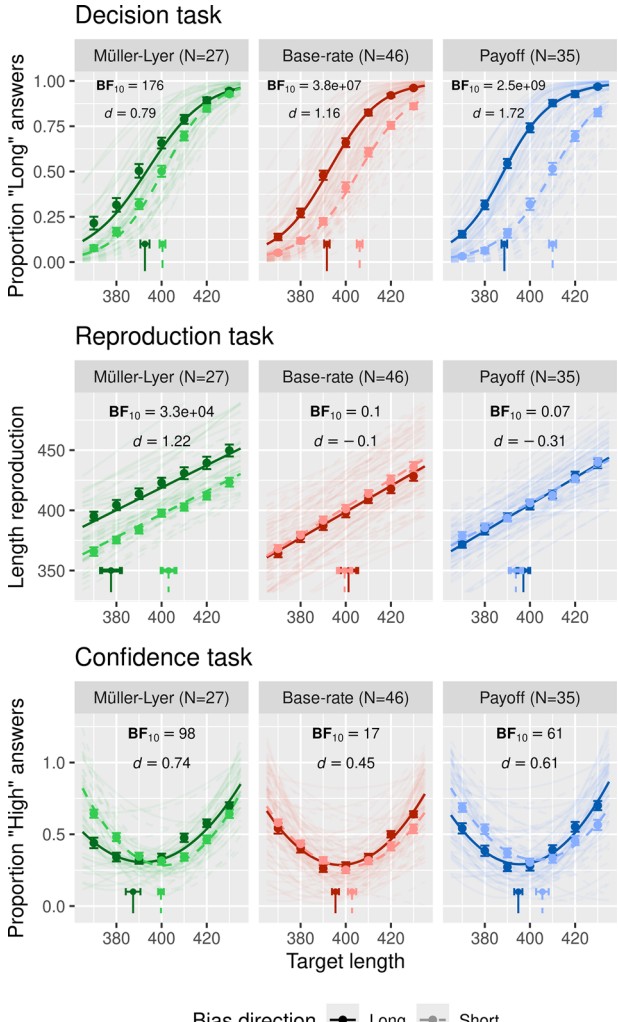

**Fig. 2 | Decision concurrent confidence results.** Top row: Decision task bias. A cumulative Gaussian function was fitted to each observer's distribution of 'short' and 'long' answers as a function of the length of the target lines separately for each bias source and bias direction condition. The group-level point of subjective equality (PSE) is plotted at the bottom of each panel. Middle row: Length reproduction. Same as described for the decision task bias except that a straight line was fitted to the distribution of length reproductions. At the bottom of each panel is plotted the target length associated with reproductions equal to the reference line (400 pixels) at the group-level. Bottom row: Confidence bias. Same as described for the decision task bias except that a quadratic function was fitted to the distribution of 'high' and 'low' confidence responses. The group-level point of peak uncertainty (lowest confidence rating) is plotted at the bottom of each panel. Across all plots dots and error bars correspond to empirical data along with fitted functions plotted as lines. All error bars indicate one standard error from the mean. Conditions biased to long are plotted as dark red, green or blue, while biased to short conditions are plotted as light red, green or blue. BF values correspond to one-sided t-tests with a default Cauchy prior of 0.707. d values correspond to Cohen's d effect size coefficients. Each panel (top, middle and bottom) depicts the data of 27 participants in the Müller–Lyer condition, 46 participants in the base rate condition and 35 participants in the Payoff condition, summing up to 108 participants.

## Reporting summary

Further information on research design is available in the Nature Portfolio Reporting Summary linked to this article.

## Results

### Experiment 1: Simultaneous decision and confidence rating

Across all conditions observers were able to distinguish between short and long lines (see Supplementary Fig. S7 for sensitivity data for each condition).

To assess bias in the decision task, we fitted a cumulative Gaussian function to each participant's distribution of 'short' and 'long' responses as a function of the length of the target lines, separately for each bias source and bias direction condition (see Fig. 2 top row). We then approximated the point of subjective equality (PSE), that is, the target length at which observers are equally likely to answer 'short' or 'long'. Overall, all manipulations resulted in large effects such that observers preferred the biased choice ('short' or 'long' depending on the bias direction). A simple paired Bayesian t-test showed extreme evidence for a difference between the biased-to-long and biased-to-short conditions in all bias manipulations ($BF_{10} = 176$, $d = 0.79$, 95% CrI = [3.3–11] in the Müller–Lyer condition; $BF_{10} = 3.8e + 07$, $d = 1.16$, 95% CrI = [10.4–17.7] in the base rate condition; $BF_{10} = 2.5e + 09$, $d = 1.72$, 95% CrI = [16.4–25] in the payoff condition; all t-tests reported are one-sided and have a default Cauchy prior of 0.707 unless stated otherwise; for all t-tests data was assumed to be distributed normally but this was not formally tested; d values correspond to Cohen's d, and CrI indicate the 95% high density credible interval of the difference between biased-to-long and biased-to-short conditions). These effects replicate our previous findings that all bias manipulations result in a response shift towards the biased option[4].

Next, to assess whether these decision shifts resulted from changes in perception we fitted a straight line to observers' distributions of length reproductions as a function of the length of the target lines, separately for each bias source and bias direction condition (see Fig. 2 middle row). We then estimated the target line (x axis) for which observers reproduced the length of the reference line (y axis). While the Müller–Lyer condition showed a large effect reflecting that biased-to-long lines were reproduced as longer and biased-to-short were reproduced as shorter, the reproduction error magnitudes in the base rate and payoff conditions were nearly identical for the biased-to-short and biased-to-long conditions. A paired Bayesian t-test revealed extreme evidence for an effect in the Müller–Lyer condition ($BF_{10} = 3.3e + 04$, $d = 1.22$, 95% CrI = [16–33.2]), and substantial evidence for a null effect in the base rate and payoff conditions ($BF_{10} = 0.1$, $d = -0.1$, 95% CrI = [−6.2–3.1] and $BF_{10} = 0.07$, $d = -0.31$, 95% CrI = [−6.4—0.6], respectively; see Supplementary Fig. S8 for the decision and reproduction results plotted as in[4]). Thus, as in Sánchez-Fuenzalida et al.[4], we show that despite the fact that the bias effect for payoff and base rate are largest compared to Müller–Lyer, reproduction effects uniquely occur in Müller–Lyer and not in payoff or base rate, confirming that only the Müller–Lyer manipulation causes a change in subjective experience.

Finally, we assessed whether confidence reports were able to distinguish between perceptual and non-perceptual manipulations as identified by the reproduction task. We reasoned that if confidence would uniquely capture changes in subjective experience, the bias manipulations towards 'long' and 'short' should translate into confidence shifts only for the Müller–Lyer, but not for the other bias manipulations. To assess shifts in confidence, we fitted a quadratic function to observers' distribution of 'low' and 'high' confidence responses as a function of the length of the target lines, separately for each bias source and bias direction condition (see Fig. 2 bottom row). We followed Gallagher et al.[15] and approximated the bottom point of each fitted curve to estimate the target length associated with the point of maximal uncertainty. All manipulations resulted in medium to large sized effects and a paired Bayesian t-test revealed at least strong evidence for a difference in confidence between bias-to-long and bias-to-short in all bias manipulations ($BF_{10} = 98$, $d = 0.74$, 95% CrI = [5.2–18] in the Müller–Lyer condition; $BF_{10} = 17$, $d = 0.45$, 95% CrI = [2.2–11.5] in the base rate condition; $BF_{10} = 61$, $d = 0.61$, 95% CrI = [4.1–15.8] in the payoff condition). That is, when the bias direction was 'long', observers were most uncertain when the length of the target line was slightly shorter than the reference line, and vice versa when the bias direction was 'short'.

Taken together, these data replicate our previous results by showing that only the Müller–Lyer illusion had an effect both on observers' decisions (decision task) and subjective experience (controlled reproduction task), whereas the payoff and base rate manipulations affected observers' decisions without changing perception. Crucially, however, we further show that all

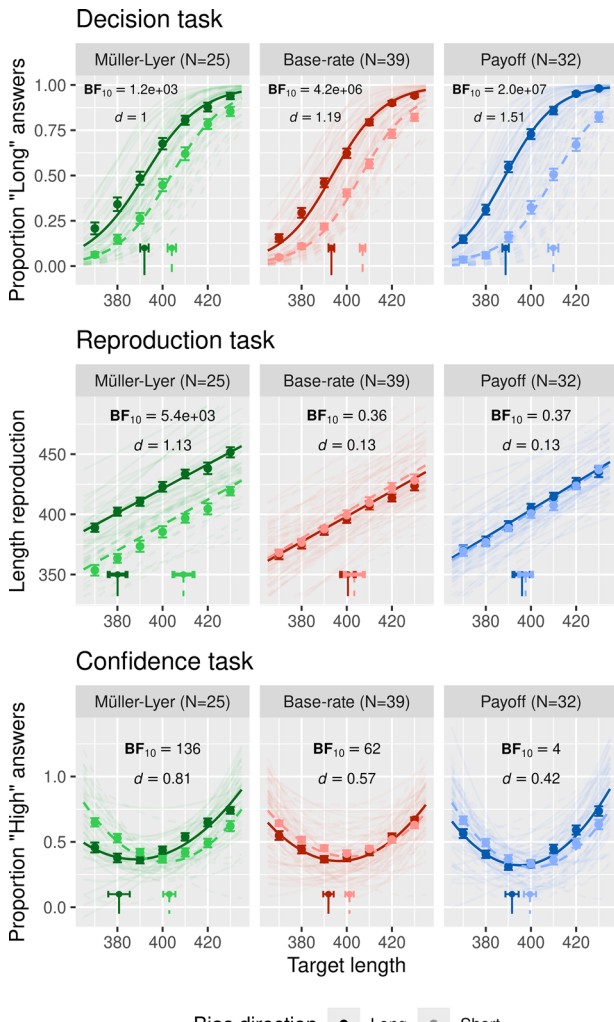

**Fig. 3 | Post-decision-confidence results.** Decision task bias (top row), length reproduction (middle row) and confidence bias (bottom row) are depicted in the same way as described in Fig. 2. Each panel (top, middle and bottom) depicts the data of 25 participants in the Müller–Lyer condition, 39 participants in the base rate condition and 32 participants in the Payoff condition, summing up to 96 participants.

three manipulations resulted in biased confidence reports, regardless of whether they affected perceptual experience.

## Experiment 2: post-decision-confidence rating

One possible explanation for the results of the previous experiment is that observers may have conflated their preference for the biased choice with their confidence reports, such that the act of choosing the biased option contaminated simultaneous ratings of confidence. To control for such effects we ran a second experiment using the same experimental design with the exception that observers first answered whether the target lines were shorter or longer than the reference and only afterwards had to report their confidence on the decision they just made. For all three tasks we followed the same analysis procedure as described in the previous experiment.

Across all conditions observers were able to distinguish between short and long lines (see Supplementary Fig. S9 for each condition's sensitivity data) and all manipulations resulted in large effects and extreme evidence for a difference between bias direction conditions ($BF_{10} = 1.2e + 03$, $d = 1$, $CrI = [6.1–16.2]$ in the Müller–Lyer condition; $BF_{10} = 4.2e + 06$, $d = 1.19$, 95% CrI = [9.4–16.9] in the base rate condition; $BF_{10} = 2.0e + 07$, $d = 1.51$, 95% CrI = [15.7–26.1] in the payoff condition; see Fig. 3, top row). In the reproduction task we again found extreme evidence for an effect in the

Müller–Lyer condition ($BF_{10} = 5.4e + 03$, $d = 1.13$, 95% CrI = [17.4–38.4]), and anecdotal evidence for a null effect in the base rate ($BF_{10} = 0.36$, $d = 0.13$, CrI = [−4.2–9.3]) and payoff ($BF_{10} = 0.37$, $d = 0.13$, CrI = [−2.7–5.5]) condition (see Fig. 3, middle row; see Supplementary Fig. S10 for the decision and reproduction results plotted as in[4]). When combining the reproduction data from experiment 1 and 2 (for which the stimuli and the task were identical), the evidence is even more compelling with a $BF_{10}$ of 7.0e + 08 for Müller–Lyer (extreme evidence for an effect, $d = 1.17$, 95% CrI = [20.4–33.3]), a $BF_{10}$ of 0.14 ($d = 0.02$, 95% CrI = [−3.8–4.3]) for base rate and a $BF_{10}$ of 0.08 ($d = −0.09$, 95% CrI = [−3.7–1.7]) for payoff (moderate and strong evidence for the null) (see Supplementary Fig. S11 for combined reproduction data). Crucially however, when testing for a difference in the point of maximal uncertainty based on the confidence responses, we find compelling evidence for effects in all conditions: extreme evidence in the Müller–Lyer condition ($BF_{10} = 136$, $d = 0.81$, 95% CrI = [8.5–31.5]), very strong evidence in the base rate condition ($BF_{10} = 62$, $d = 0.57$, 95% CrI = [3.4–13.9]) and substantial evidence in the payoff condition ($BF_{10} = 4$, $d = 0.42$, 95% CrI = [0.6–13.8]) (see Fig. 3, bottom row). This confirms the findings from experiment 1, that confidence responses do not uniquely shift as a result of changes in perceptual experience, but also as a result of non-perceptual information, even if such information does not aid decision accuracy.

Thus, on the one hand, the results of the combined data of both experiments confirm that the reproduction task consistently tracks the effect of the Müller–Lyer manipulation while being unaffected by the payoff and base rate manipulations. On the other hand, confidence reports were biased across all conditions–both when confidence reports were elicited simultaneously with a decision, and also when confidence was rated after the decision.

## Ordinal modelling

Overall, the effects described for both experiments are consistent with all three bias manipulations affecting both the criterion (decision task) and confidence. Conversely, both experiments show that the Müller–Lyer illusion selectively affects subjective experience while the payoff and base rate manipulations do not (as evidenced by the combined reproduction data of Experiments 1 and 2). To statistically underpin this pattern of results, we combined the data of both experiments and adopted a Bayesian model comparison framework to test for ordinal-constrained models (see[27] for an in-depth explanation of the method and[35] for a practical application). This statistical framework allows one to translate concrete, ordinal constellations of effects into statistical models that can be compared directly by computing their relative likelihood (see Analysis—Bayesian Model Comparison). We created a full set of models specifying all the possible combinations of presence and absence of effects across the three bias manipulations (see Supplementary Fig. S6 for the specification of all models). Both in the decision ($BF_{A\text{-over-null}} = 1.2e + 52$) and in the confidence ($BF_{A\text{-over-null}} = 8.2e + 11$) task, the best performing model (A) described a bias effect across all conditions, whereas in the reproduction task, the best performing model (C) was one in which the Müller–Lyer had an effect, while the base rate and payoff had null effects ($BF_{C\text{-over-null}} = 1.4e + 18$; see Fig. 4). In all tasks, the best performing model also beat the second-best performing model ($BF_{A\text{-over-E}} = 2.2e + 06$ in the decision task; $BF_{A\text{-over-F}} = 370$ in the confidence task; $BF_{A\text{-over-F}} = 4.9$ in the reproduction task). Taken together this analysis confirms that all three manipulations affect decisions and confidence reports, whereas only the Müller–Lyer illusion affects subjective experience, as measured by the reproduction task.

## Decision and confidence effects

Although all bias manipulations affected first-order decisions and confidence reports, we conducted an exploratory analysis to assess whether these two measures displayed differential sensitivity to perceptual and non-perceptual manipulations. We combined the data of experiments 1 and 2 to calculate the effect of each manipulation (Müller–Lyer, base rate and payoff) on the point of subjective equality (decision task) or the point of maximal

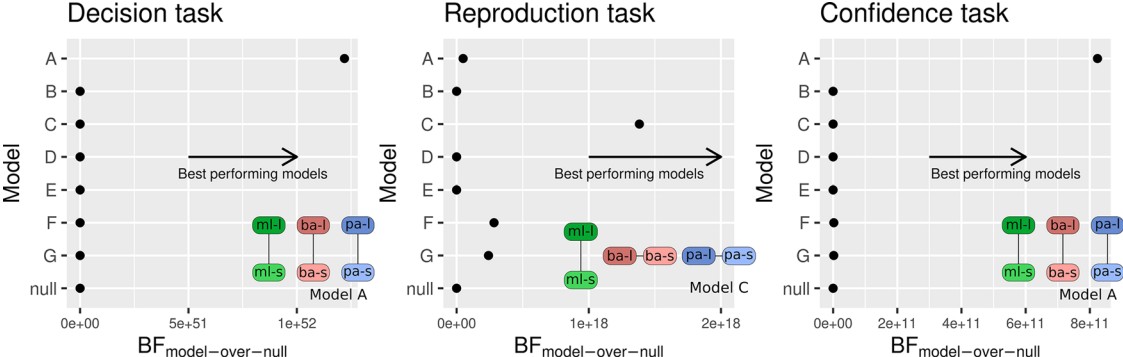

**Fig. 4 | Ordinal model comparison.** Bayes factor values for each of the ordinal models tested. All models were compared against a baseline model (null model). In the null model, there was no effect between bias-to-long and bias-to-short for a given bias source condition, but there could be differences between bias sources. Higher values of $BF_{model-over-null}$ indicate a better performance of a given model on the y-axis in comparison with the baseline model (null model). For each task, a graphical depiction of the winning model is included (model A in the decision and confidence task, and model C in the reproduction task), where ml stands for Müller–Lyer, ba for base rate, and pa for payoff, l for biased-to-long, and s for biased-to-short. For a graphical depiction of all the models, see Supplementary Fig. S6.

uncertainty (confidence task) across the bias-to-long and bias-to-short conditions. This metric provided us with an index of the strength of the effect each manipulation had on each measure for each participant (see Fig. 5 Top). Using ordinal modelling, as described in the previous section, we compared a new set of models to determine, for a given condition, whether the effect on the decision task was equal, smaller or larger than the effect on confidence (see Supplementary Fig. S12). While the Müller–Lyer illusion had a larger effect on confidence reports ($d = 0.42$, 95% CrI = [2.3–11.4]; credible intervals (CrI) refer to the difference between the effect on the decision task and confidence task), the base rate ($d = -0.41$, 95% CrI = [−8.6 to −2.7]) and payoff ($d = -0.62$, 95% CrI = [−16.1 to −7.2]) manipulations had a bigger effect on first-order decisions ($BF_{AD-over-null} = 2.9e+08$; $BF_{AD-over-second-best-model} = 30$; see Fig. 5 Bottom for a graphical depiction of the best-performing model and Supplementary Fig. S13 for the BF values of all the models tested). Note that the results are similar when each experiment is analysed separately (see Supplementary Fig. S14). To summarise, this analysis suggests that perceptual and non-perceptual manipulations (as identified by the reproduction task) load differentially on first-order decisions and confidence reports.

**Exploratory analysis of metacognitive efficiency**

Finally, we performed an exploratory analysis of the effect of our experimental manipulations on metacognition–the mapping between confidence and performance over many trials. Metacognitive sensitivity is often quantified using meta-d', a measure that computes the type-1 sensitivity (d') one would expect from the observer's metacognitive (type 2) performance, on the assumption that the observer is metacognitively ideal[31]. To control for first-order effects of the manipulations, we computed metacognitive efficiency, reflecting the metacognitive performance of the observer controlling for their type-1 sensitivity (meta-d'/d', or M-ratio, see Methods[10,31]);.

Given that the manipulations were symmetrical (one condition biased line length in one direction while the other biased in the other direction) and that we controlled for first-order effects on d', we expected that metacognitive efficiency should be the same for bias-to-short and bias-to-long in all manipulations. To test this, we estimated M-ratio (meta-d'/d') for each condition in both experiments (see Analysis–Metacognitive efficiency for details on the estimation of M-ratio). The average M-ratio was lower than 1 in all conditions, as is typical in studies of metacognition of perceptual decisions (see Supplementary Fig. S15 for the posterior distributions of M-ratio). When confidence was reported concurrently with the decision (Experiment 1) there was moderate evidence for no effect on M-ratio between bias-to-long and bias-to-short in the Müller–Lyer condition ($BF_{10} = 0.28, d = 0.03$, 95% CrI = [−0.03–0.03]; BF values are obtained using the Savage-Dickey density ratio, see Methods for a detailed account; credible

intervals (CrI) refer to the difference between the biased-to-long and biased-to-short M-ratio). Surprisingly, however, we observed decisive evidence for an effect of bias on M-ratio in the base-rate condition ($BF_{10} = 507, d = 0.99$, 95% CrI = [0.25–0.22]) and anecdotal evidence for an effect of bias on M-ratio in the payoff condition ($BF_{10} = 1.58, d = 0.53$, 95% CrI = [0.18–0.10]). When confidence was reported after the decision had been made (Experiment 2), there was again moderate evidence for no effect in the Müller–Lyer condition ($BF_{10} = 0.29, d = 0.19$, 95% CrI = [0.05–0.07]), anecdotal evidence for an effect in the base rate condition ($BF_{10} = 0.76$, $d = 0.39$, 95% CrI = [0.12–0.15]), and again anecdotal evidence for an effect in the payoff condition ($BF_{10} = 0.96, d = 0.45$, 95% CrI = [0.09–0.11]). The direction of these effects was always such that bias-to-long was associated with higher metacognitive efficiency compared to bias-to-short. Summarising these results, non-perceptual manipulations provide anecdotal to strong evidence for higher metacognitive efficiency for bias-to-long compared to bias-to-short, while the Müller–Lyer manipulation provides moderate evidence for no effect of bias on metacognitive efficiency.

**Discussion**

Across two experiments using three well-known bias manipulations (the Müller–Lyer illusion, a base rate manipulation and a payoff manipulation) we showed that confidence reports in a perceptual decision-making context are susceptible to bias regardless of the perceptual or non-perceptual nature of the manipulations, and even when these manipulations do not affect measures of subjective experience. First, we replicated our previous results[4] showing that the Müller–Lyer illusion biases first order decisions as well as subjective experience, whereas base rate and payoff manipulations selectively biased decisions without affecting subjective experience. We then showed that all bias manipulations resulted in biased confidence reports when reported concurrently or after the first-order decision. However, when taken together, the relative bias effects on first-order decisions and on confidence reports offered differential outcomes for perceptual and non-perceptual manipulations. This demonstrates that, although confidence in a perceptual task may partially reflect subjective experience, it is also influenced by non-perceptual information.

Confidence has been conceptualised as emerging during the decision-making process[36–40], with additional processing occurring post-decisionally, after a judgement has been made[41,42]. Under inferential models of confidence formation, a number of cues—such as response time, or motor fluency—may be leveraged to inform a summary judgement of the likelihood of a decision being correct[37]. To investigate how biases influence these different aspects of confidence formation we asked observers to either report their confidence concurrently with their decision (experiment 1) or report confidence after the decision had been made (experiment 2). Although post-decisional confidence reports are usually considered to be more affected by

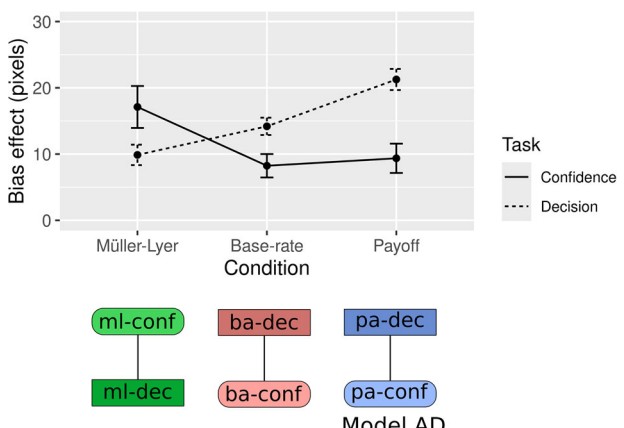

**Fig. 5 | Effects on the decision and confidence task.** For each condition and task, the difference in target length (bias-to-long minus bias-to-short) is associated with the point of subjective equality (decision task) or the point of maximal uncertainty (confidence task). Error bars indicate the SEM. The best performing model (AD) is depicted below the graph, where "conf" indicates the bias effect in the confidence task and "dec" indicates the bias effect in the decision task for the Müller–Lyer ("ml"), base rate ("ba") and payoff ("pa"). Each point in the Müller–Lyer condition corresponds to the data of 52 participants, 85 participants in the base rate condition and 67 in the payoff condition, summing up to 204 participants.

non-perceptual information[42–44], we found that both concurrent and post-decision-confidence reports were biased across all manipulations.

Both perceptual (Müller–Lyer illusion) and non-perceptual (base rate and payoff) manipulations, as identified by the reproduction task (combined data of Experiments 1 and 2), resulted in biased confidence reports regardless of whether the confidence report was concurrent or delayed. As the Müller–Lyer illusion affects perceptual experience, the effect of this manipulation on confidence reports is in line with Gallagher and colleagues' (2019)[15] findings, as well as other research that shows concomitant effects in decision and confidence when using manipulations which are assumed to be perceptual in nature[45–50]. However, despite not affecting perception, the base rate and payoff manipulations also resulted in biased confidence reports. Indeed, it has been argued before[6], and shown experimentally[16,20,51–55], that observers can use prior information to inform their confidence judgments. While this behaviour can be considered normatively optimal, as prior information can be used to influence decision performance, and therefore confidence, it has also been shown that false prior information[56], as well as the rewards associated with a decision[20,21], can affect confidence even if they have no bearing on decision accuracy (see ref. 49 for an example in which rewards did not affect confidence reports). Importantly though, previous studies have not established whether these effects of non-perceptual manipulations on confidence are also accompanied by changes in subjective experience.

Here we addressed this question by establishing that confidence reports are biased by both base rate and expected rewards while also confirming that these manipulations did not affect subjective experience as measured through the reproduction task (combined data of Experiments 1 and 2). Nevertheless, when evaluated together, first-order decisions and confidence reports offered differential outcomes for perceptual and non-perceptual manipulations. While the Müller–Lyer illusion had a larger effect on confidence reports, the base rate and payoff conditions had a larger effect on first-order decisions. In line with these results, previous research has shown reduced effects on confidence compared to first-order decisions[45,46], which have been taken as indicating that perceptual decisions are subject to an additional source of bias or that a different decision criterion is employed for confidence responses (Arnold et al.[57]). However, it is worth noting that the reduced effect of non-perceptual biases on confidence found in our study as well as others could indicate that perceptual decisions and confidence reports differ in sensitivity to bias[58,59]. Further, we also found that the

Müller–Lyer illusion more strongly biased confidence reports than first-order decisions. This interaction suggests that, although confidence reports were influenced by non-perceptual manipulations, they are still more sensitive to perceptual manipulations. Alternatively, the relatively stronger effect on confidence compared to first-order decisions in the Müller–Lyer illusion condition could also have resulted from the fact that participants have prior information about the Müller–Lyer illusion (due to the fact that many of them are psychology students). As a result, they may have attempted to actively counteract the effect of the illusion in the decision task, thereby compensating for its effect on their decisions, while failing to do so when asked for confidence estimates[50]. Although this account is speculative, it could explain why the biasing effect on first-order decisions in the Müller–Lyer condition is smaller than the biasing effects on confidence, which would be hard to explain otherwise.

Our results reveal a dissociation between confidence reports and measures of subjective experience, with confidence tracking both subjective experience and other non-perceptual information. Such results are consistent with computational models that propose confidence tracks choice consistency rather than perceptual experience per se[58,60]. Our findings also echo previous literature that have warned about the dangers of equating confidence level alone with subjective experience[10,11,61]. However, in line with previous research our results also suggest that confidence can be used to index effects in subjective experience, as long as care is taken to account for the possible influence of non-perceptual factors[15,45–49,56]. Notably, we found perceptual and non-perceptual manipulations could be distinguished when evaluating their relative effects on first- and second-order decisions. It is not yet fully clear what mechanism supports this interaction or whether these results will extend to other perceptual and non-perceptual manipulations. However, our results do suggest a more nuanced appraisal of effects in which perceptual manipulations are assessed by their relative influence on decisions and confidence rather than supporting all-or-nothing claims.

Our data revealed strong effects of all bias manipulations on raw confidence scores. As we argue above, this casts doubt on a one-to-one mapping between confidence and subjective experience. An alternative perspective suggests that subjective experience is correlated with aggregate scores that reflect metacognitive sensitivity (the ability to distinguish correct from incorrect decisions using confidence) or metacognitive efficiency (metacognitive sensitivity normalised by type I performance) computed over many trials (see[12] for arguments in favour of this approach in the context of detection tasks commonly used in consciousness research). To investigate how such measures fare in the context of perceptual and non-perceptual biases, we explored metrics of metacognitive efficiency in our tasks (M-ratio[10,31]). It is important to note here that we expected on theoretical grounds that there should be no M-ratio difference between bias-to-long and bias-to short in any of the conditions, because there is no a-priori reason to expect that metacognitive efficiency should be different for bias-to-long versus bias-to-short. Surprisingly, there was a large effect in metacognitive efficiency in the base rate condition in Experiment 1, such that the bias-to-short condition showed reduced efficiency as compared to the bias-to-long condition. Additionally, there was moderate evidence for no effect in the Müller–Lyer condition in both experiments. Although the other conditions provided insufficient evidence to convincingly argue for an effect or lack thereof, these results suggests that non-perceptual effects on confidence may also indeed leak into measures of metacognitive efficiency, especially when caused by differences in base rate. While the overall meaning of this pattern of results remains somewhat elusive, especially as there is no a-priori reason to expect differences in metacognitive efficiency for bias-to-long versus bias-to-short, future research may wish to explore how measures of metacognitive insight could distinguish between perceptual and non-perceptual biases (for example, see refs. 50,62).

Regardless of how one interprets these results, it is important to point out that any current decision-making model requires one to set a ground truth for what the observer 'should' detect, or what the

experience of the observer 'should' be according to the experimenter (the experimenter's ground truth). The act of setting this ground truth forces a reality onto the observer that may not align with their actual experience (an editorial highlighting this issue can be found in ref. [63]). For example, the observed 'biases' in the Müller–Lyer manipulation can best be understood as cases in which the experimenter sets a ground truth that is different from what the observer actually experiences. This problem lies at the heart of the article by Witt et al.[3], in which they showed that SDT cannot dissociate perceptual from post-perceptual effects (also see section 6.11 in refs. [4,12]). A framework that sets a ground truth will intrinsically have trouble distinguishing between bias effects that are attributable to the way the experimenter sets the ground truth, or to actual response biases by the observer. Thus, due to the requirement of having to set a principally unknowable ground truth, attempting to use metacognitive or other signal detection theoretic measures to dissociate effects on subjective experience from post-perceptual effects might be considered putting the cart before the horse. Similarly, while confidence ratings are invaluable for gaining insight into the metacognitive profile of an observer, they must be used with caution when attempting to draw conclusions about subjective experience itself, as our data illustrate.

Indeed, establishing whether a given manipulation affects subjective experience requires an independent benchmark, as provided here by our controlled reproduction measure. Importantly, it is intrinsically impossible to establish with certainty whether any given measure–including reproduction–uniquely assays the subjective nature of an effect, or whether it (also) reflects non-perceptual decisional or response biases[1]. Reproduction tasks have themselves previously been shown to be biased by non-perceptual effects, as caused by trial history or other contextual factors[64–66]. Thus, one may reasonably also question whether our controlled reproduction task uniquely captures subjective experience in the current study. Note, however, that a critical feature of the current results is that when combining the data of Experiments 1 and 2, the controlled reproduction task produced null effects in two out of three conditions (base rate and payoff). It is unlikely that non-perceptual effects would have acted to perfectly cancel out changes in perceptual effects in those conditions, and thus a null result can be securely interpreted as evidence for a lack of change in subjective experience.

## Limitations

A limitation of our study is that we investigated a restricted set of manipulations. Although the effects of payoff and base rate are similar on the measures for which we collected data, other bias types might produce different results. Similarly, only one of our manipulations (the Müller–Lyer) seemed to result in perceptual biases, and whether other perceptual manipulations would produce similar effects remains to be discovered in future research.

## Conclusion

In conclusion, our findings show that although confidence can be used to interrogate perception, confidence reports can also dissociate from subjective experience in perceptual decision-making. Thus, investigating changes in subjective experience in psychophysics, metacognition, and consciousness research remains an extremely challenging task and will not be accomplished using a one-size-fits-all approach.

## Data availability

All data presented and analysed in this paper are publicly available at https://osf.io/h3uj5/.

## Code availability

All the scripts used in the analysis as well as the code to generate the plots are publicly available at https://osf.io/h3uj5/.

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

## Acknowledgements

N.S.F. is funded by Agencia Nacional de Investigación y Desarrollo (ANID; 72190272); N.S.F., S.v.G, and J.J.F. are funded by H2020 European Research Council (ERC) PRIORITY Excellent science (ERC-2016-STG_715605); S.M.F. is a CIFAR Fellow in the Brain, Mind & Consciousness Programme and funded by UKRI under the UK government's Horizon Europe funding guarantee [selected as ERC Consolidator, grant number 101043666]. The funders had no role in study design, data collection and analysis, decision to publish, or preparation of the manuscript.

## Author contributions

N.S.F.: conceptualisation, funding acquisition, investigation, formal analysis, methodology, project administration, software, validation, visualisation, writing–original draft. S.v.G.: conceptualisation, funding acquisition, methodology, supervision, writing–review & editing. S.M.F.: conceptualisation, writing–review & editing. J.M.H.: methodology, resources, writing–review & editing. J.J.F.: conceptualisation, methodology, supervision, project administration, writing–review & editing.

## Competing interests

The authors declare no competing interests.
