## [Transparent Peer Review file · Communications Psychology]

Confidence reports during perceptual decision making dissociate from changes in subjective experience

Corresponding Author: Dr Nicolás Sánchez-Fuenzalida

Version 0:

Decision Letter:

Dear Dr Sánchez-Fuenzalida,

Thank you for your patience during the peer-review process. We are sorry for the exceptionally long delay. Your manuscript titled "Confidence reports during perceptual decision making dissociate from changes in subjective experience" has now been seen by 2 experts in the field, and I include their comments at the end of this message. They find your work of interest but raised some important points. We are interested in the possibility of publishing your study in Communications Psychology, but would like to consider your responses to these concerns and assess a revised manuscript before we make a final decision on publication.

We therefore invite you to revise and resubmit your manuscript, along with a point-by-point response to the reviewers. Please highlight all changes in the manuscript text file.

Editorially, we consider it important that the revised manuscript includes the test of equivalent bias estimates between the analyses of confidence and of perceptual decisions, as suggested by Reviewer #1. In addition, please clarify the different bias estimates and how they may deliver similar or different outcomes. Please refer to the attached checklist and the guidance on our homepage for information on our policies for data and code sharing.

I am attaching an Editorial Requests Table that details critical reporting requirements for the revised manuscript. Please attend to each item and ensure your manuscript is fully compliant. If your revised manuscript is not aligned with these requests on major issues, such as those concerning statistics, it may be returned to you for further revisions without re-review.

Please submit the following items:

- Revised manuscript
- Point-by-point response to the referees' comments
- Cover letter (as a separate document)
- [Nature Research Reporting Summary](https://www.nature.com/documents/nr-reporting-summary.zip)
- [Editorial Policy Checklist](https://www.nature.com/documents/nr-editorial-policy-checklist.pdf)
- Completed Editorial Request Table (attached).

via this link: Link Redacted .

Additional guidance is available in our style and formatting guide Communications Psychology formatting guide.

Best regards,

Troby Lui

Troby Lui, PhD
Associate Editor
Communications Psychology

REVIEWER EXPERTISE:

Reviewer #1: metacognition, signal detection theory

Reviewer #2: metacognition, signal detection theory, meta-cognitive sensitivity

REVIEWER REPORTS:

Reviewer #1 (Remarks to the Author):

Overall comments:

I am generally positive about this submission, as I think there are some useful data here that could benefit the field. Clearly in this context reproductions have been less susceptible to the biasing influence of base rate manipulations and differential payoff.

However, at the moment I fear the potential positive contributions this manuscript could make are being undermined by a treatment of data that lacks nuance.

I outline my major points below.

Major Points

1) The authors need to test for equivalent bias estimates across conditions. Are the bias estimates delivered by analyses of confidence (peak uncertainty shift estimates) reduced relative to those delivered by perceptual decisions (PSE shift estimates)?

It is dangerous to eyeball data, as this fails to capture within participant effects, but in Figures 2 and 3 the bias estimates delivered by analyses of confidence seem to be markedly reduced relative to estimates delivered by analyses of perceptual decisions. This would imply that these two types of analysis have delivered dissociable outcomes, and that perceptual decisions have either been contaminated by an additional biasing factor, or they have been more influenced by a common factor.

2) If I am correct, and analyses of confidence and perceptual decisions are indeed delivering distinct outcomes, then the modelling would have to be updated to account for this, and the discussion would have to be re-written.

3) Even if I am mistaken, and analyses of confidence and of perceptual decisions have delivered inseparable outcomes in

these sets of data, the Discussion should still be substantially updated to discuss the numerous occasions when analyses of confidence, and of decision speeds (which are directly interrelated), have delivered separable outcomes.

Have all of the relevant authors simply been wrong? Or can these different analyses sometimes deliver different outcomes? If they can, the authors should speculate about what conditions could cause these different analyses to deliver different outcomes, and that should encompass an account of why their own manipulations have instead delivered more similar outcomes.

Some examples...

Bruno, A., Sudkamp, J., & Souto, D. (2023). A metacognitive approach to the study of motion-induced duration biases reveals inter-individual differences in forming confidence judgments. *Journal of Vision*, 23(3), 15-15.

Gallagher, R. M., Suddendorf, T., & Arnold, D. H. (2021). The implied motion aftereffect changes decisions, but not confidence. *Attention, Perception, & Psychophysics*, 83(8), 3047-3055.

Bouyer, L. N., Arnold, D. H., Johnston, A., & Taubert, J. (2023). Predictive extrapolation effects can have a greater impact on visual decisions, while visual adaptation has a greater impact on conscious visual experience. *Consciousness and Cognition*, 115, 103583.

Luna, R., Serrano-Pedraza, I., Gegenfurtner, K. R., Schütz, A. C., & Souto, D. (2021). Achieving visual stability during smooth pursuit eye movements: Directional and confidence judgements favor a recalibration model. *Vision research*, 184, 58-73.

Mihali, A., Broeker, M., Ragalmuto, F. D., & Horga, G. (2023). Introspective inference counteracts perceptual distortion. *Nature Communications*, 14(1), 7826.

Maldonado Moscoso PA, Cicchini GM, Arrighi R, Burr DC. Adaptation to hand-tapping affects sensory processing of numerosity directly: evidence from reaction times and confidence. *Proc Biol Sci*. 2020 May 27;287(1927):20200801. doi: 10.1098/rspb.2020.0801

4) If I am correct, and analyses of confidence and of perceptual decisions have indeed delivered different outcomes, that would in no way undermine the fact that in this case reproductions have delivered a different set of outcomes again, with results that are consistent with an absence of bias across the two critical conditions, whereas the other two types of analysis are both indicative (possibly of different magnitudes) of bias.

That would be interesting. However, the discussion would then be about why reproductions have been even less prone to bias than confidence (or presumably response time) based analyses – as opposed to the current treatment which asserts that confidence based analyses are inseparable from perceptual decisions. I think this treatment lacks nuance.

5) The authors should make their data publicly available. I could have easily answered these questions myself in a few minutes if they had.

6) Are all biases the same, or will different types of bias have a differential impact on different measures?

Here the authors have chosen to bias participants via manipulating base rates and pay off rates. Will these types of bias deliver the same results as say asking people to judge a room's brightness after pondering their misdeeds, or after pondering past kind acts?

I would like to see some discussion regarding this general type of issue.

7) Will reproductions always be possible?

Reproductions have always been a popular psychophysical task (they were not newly introduced to the field in 2023), both because they minimise bias and we are often interested in the perceptual experience of features that can easily be reproduced (e.g. tilt, or colour). However, we are also often interested in qualities (perceptual and cognitive) that might not be so easily reproduced in an experimental protocol.

What would the authors recommend as a means to gain insight into the underlying cause of a bias when a reproduction protocol is untenable?

Reviewer #2 (Remarks to the Author):

By integrating the standard perceptual decision-making paradigm with the controlled reproduction method, the manuscript effectively demonstrates how first-order decision and second-order confidence ratings are influenced by perceptual and non-perceptual manipulations. The research is grounded in robust methodology, and the findings hold significant value for the field. I only have relatively minor comments, which should be addressed to further strengthen the manuscript.

1. Visual inspection suggests that the point of maximal uncertainty (PMU) exhibited a larger shift than the point of subjective equality (PSE) under the perceptual manipulation, whereas the reverse appears true in the non-perceptual manipulations. These trends were consistent across Experiments 1 and 2, prompting me to consider the potential significance of this finding in the study of consciousness. If the authors deem it relevant, I would appreciate further discussion of this observation.

2. The term "bias" is a bit overused, carrying different meanings depending on the context. For instance, "response bias" in line 49 exclusively refers to non-perceptual biases, such as those induced by base rate or payoff manipulations. Meanwhile, the "bias parameter" mentioned in line 51 pertains to the criterion in signal detection theory. Later, "biases in confidence" in line 101 describes shifts in empirical confidence data caused by whatever factors. Adopting consistent terminology would enhance readability, particularly for beginners.

3. I was not quite following the paragraph starting from line 560. In my understanding, and as the authors noted in line 50, SDT serves only to dissociate sensitivity (d') from criterion (c) in decision-making. This only indicates these two parameters are estimated independently without carrying any further meaning. It is unfortunate that some researchers misinterpreted these parameters solely reflecting perceptual and non-perceptual effects. Yet, since SDT was never intended to distinguish between perceptual and non-perceptual effects, I find it even more difficult to grasp the intention behind this paragraph.

Version 1:

Decision Letter:

Dear Dr Sánchez-Fuenzalida,

Your manuscript titled "Confidence reports during perceptual decision making dissociate from changes in subjective experience" has now been seen by our reviewers, whose comments appear below. In light of their advice I am delighted to say that we are happy, in principle, to publish a suitably revised version in Communications Psychology.

We therefore invite you to revise your paper one last time to address the remaining concerns of our reviewers and a list of editorial requests. At the same time we ask that you edit your manuscript to comply with our format requirements and to maximise the accessibility and therefore the impact of your work.

EDITORIAL REQUESTS:

Please review our specific editorial comments and requests regarding your manuscript in the attached "Editorial Requests

Table". Please outline your response to each request in the right hand column. Please upload the completed table with your manuscript files as a Related Manuscript file.

SUBMISSION INFORMATION:

OPEN ACCESS:

* DATA AVAILABILITY:

Link Redacted

Best regards,

Troy Lui

Troy Lui, PhD
Associate Editor
Communications Psychology

REVIEWERS' COMMENTS:

Reviewer #1 (Remarks to the Author):

I would like to thank the authors for such a positive response to the review process. I think this has resulted in a more balanced manuscript that now poses further interesting questions - while making an important immediate contribution to the field.

I felt that some of the modelling was unnecessarily opaque, whilst on my appraisal being entirely correct and sensible. I do not have any major or minor comments, other than to say well done.

Reviewer #2 (Remarks to the Author):

The authors have thoroughly addressed all my comments, and I believe the manuscript is ready for publication.

Dear editor,

Thank you for your positive response. We have carefully revised the manuscript according to the comments and suggestions provided by the reviewers. We believe we have incorporated all the suggestions into the manuscript.

In particular, we now provide a figure and analysis that directly test the effects of the manipulations over first-order decisions and confidence reports as suggested by Reviewers #1 and #2. The results of this analysis show that while the effect of the Müller-Lyer illusion is stronger on confidence, non-perceptual manipulations (base rate and payoff) have a stronger effect over first-order decisions, suggesting decisions and confidence reports are differentially affected by perceptual and non-perceptual manipulations. We have added a new section to the results (Decision and confidence effects on page 14, line 477) and we reworked the discussion and conclusion to account for these results.

Further, we address several other remarks brought up by the reviewers relating to the use of potentially confusing terms (e.g. response bias/decision bias, etc.), the potential effect of other bias manipulations and the feasibility of the reproduction task in other contexts.

Below we provide a document with detailed point-by-point responses to reviewers' comments, referring to the locations in the manuscript where changes have been made. For convenience, we also provide a document that highlights all the changes that were made relative to the original submission (old_new.pdf).

Thank you for your time and consideration, we hope the revision now meets the requirements for publications in Nature Communications Psychology.

Sincerely,

Nicolás Sánchez-Fuenzalida
Simon van Gaal
Stephen M. Fleming
Julia M. Haaf
Johannes J. Fahrenfort

Review comments

Reviewer #1 (Remarks to the Author)

Overall comments:

I am generally positive about this submission, as I think there are some useful data here that could benefit the field. Clearly in this context reproductions have been less susceptible to the biasing influence of base rate manipulations and differential payoff.

However, at the moment I fear the potential positive contributions this manuscript could make are being undermined by a treatment of data that lacks nuance.

I outline my major points below.

Major Points

- 1) The authors need to test for equivalent bias estimates across conditions. Are the bias estimates delivered by analyses of confidence (peak uncertainty shift estimates) reduced relative to those delivered by perceptual decisions (PSE shift estimates)?

It is dangerous to eyeball data, as this fails to capture within participant effects, but in Figures 2 and 3 the bias estimates delivered by analyses of confidence seem to be markedly reduced relative to estimates delivered by analyses of perceptual decisions. This would imply that these two types of analysis have delivered dissociable outcomes, and that perceptual decisions have either been contaminated by an additional biasing factor, or they have been more influenced by a common factor.

- 2) If I am correct, and analyses of confidence and perceptual decisions are indeed delivering distinct outcomes, then the modelling would have to be updated to account for this, and the discussion would have to be re-written.
- 3) Even if I am mistaken, and analyses of confidence and of perceptual decisions have delivered inseparable outcomes in these sets of data, the Discussion should still be substantially updated to discuss the numerous occasions when analyses of confidence, and of decision speeds (which are directly interrelated), have delivered separable outcomes.

Have all of the relevant authors simply been wrong? Or can these different analyses sometimes deliver different outcomes? If they can, the authors should speculate about what conditions could cause these different analyses to deliver different outcomes, and that should encompass an account of why their own manipulations have instead delivered more similar outcomes.

Some examples...

Bruno, A., Sudkamp, J., & Souto, D. (2023). A metacognitive approach to the study of motion-induced duration biases reveals inter-individual differences in forming confidence judgments. *Journal of Vision*, 23(3), 15-15.

Gallagher, R. M., Suddendorf, T., & Arnold, D. H. (2021). The implied motion aftereffect changes decisions, but not confidence. *Attention, Perception, & Psychophysics*, 83(8), 3047-3055.

Bouyer, L. N., Arnold, D. H., Johnston, A., & Taubert, J. (2023). Predictive extrapolation effects can have a greater impact on visual decisions, while visual adaptation has a greater impact on conscious visual experience. *Consciousness and Cognition*, 115, 103583.

Luna, R., Serrano-Pedraza, I., Gegenfurtner, K. R., Schütz, A. C., & Souto, D. (2021). Achieving visual stability during smooth pursuit eye movements: Directional and confidence judgements favor a recalibration model. *Vision research*, 184, 58-73.

Mihali, A., Broeker, M., Ragalmuto, F. D., & Horga, G. (2023). Introspective inference counteracts perceptual distortion. *Nature Communications*, 14(1), 7826.

Maldonado Moscoso PA, Cicchini GM, Arrighi R, Burr DC. Adaptation to hand-tapping affects sensory processing of numerosity directly: evidence from reaction times and confidence. *Proc Biol Sci*. 2020 May 27;287(1927):20200801. doi: 10.1098/rspb.2020.0801

- 4) If I am correct, and analyses of confidence and of perceptual decisions have indeed delivered different outcomes, that would in no way undermine the fact that in this case reproductions have delivered a different set of outcomes again, with results that are consistent with an absence of bias across the two critical conditions, whereas the other two types of analysis are both indicative (possibly of different magnitudes) of bias.

That would be interesting. However, the discussion would then be about why reproductions have been even less prone to bias than confidence (or presumably response time) based analyses – as opposed to the current treatment which asserts that confidence based analyses are inseparable from perceptual decisions. I think this treatment lacks nuance.

Author responses in green

We want to first thank reviewer #1 for taking the time to read our manuscript and for the many helpful comments provided. We believe the revised document is a better manuscript because of them.

Here, we address points 1) through 4), as they all refer to the same issue. Thank you for pointing out the relative strength of the effects on first-order decisions and confidence reports. Initially, we aimed to test whether confidence reports would be unaffected by non-perceptual manipulations, as suggested by Gallagher et al. (2019). In doing so, we opted not to directly test the relative strength of the manipulations over first-order decisions and confidence reports, as we did not have a concrete hypothesis about such a comparison. We understand we may have unintentionally brushed away crucial differences between first-order decisions and confidence reports, as well as previous literature showing that these measures often result in different outcomes.

To offer a more nuanced treatment of these effects, we have conducted an extra analysis that uses ordinal modelling to test whether the manipulations preferentially affected decisions or confidence. In this new analysis (Decision and confidence effects on page 14, line 477; note that line numbers refer to the revised document, not the tracked-changes document), we show that while the impact of the Müller-Lyer illusion was larger on confidence reports, the base rate and payoff manipulations had a stronger effect on first-order decisions (see newly added Figure 5). These results suggest that although confidence reports were affected by non-perceptual information, confidence and decision responses offered different outcomes for perceptual (Müller-Lyer) and non-perceptual (base rate and payoff) manipulations. We performed this analysis on the combined data of experiments 1 and 2. However, the results were similar when analysing the data of each experiment separately (see new Supplementary Figure S14).

Further, we have reworked the discussion to treat these new results. We discuss potential accounts for these results and consider the implications of using confidence reports to track subjective

experience. We also contextualise these results in light of previous literature showing differential outcomes in decisions and confidence reports.

Also note that a small error in the initial analyses came to light when we redid our analyses. This error did not change any of the results, but we did update the manuscript to fix it, see for further explanation on the last page of this rebuttal.

- 5) The authors should make their data publicly available. I could have easily answered these questions myself in a few minutes if they had.

We apologize that you did not receive the private link we created for the reviewing process. We have made the repository public instead of creating a private link for the reviewing process. The link has been added below to prevent further misunderstanding.

<https://osf.io/h3uj5/>

- 6) Are all biases the same, or will different types of bias have a differential impact on different measures?

Here the authors have chosen to bias participants via manipulating base rates and pay off rates. Will these types of bias deliver the same results as say asking people to judge a room's brightness after pondering their misdeeds, or after pondering past kind acts?

I would like to see some discussion regarding this general type of issue.

It is not possible to say which kinds of manipulations deliver which kinds of effects without testing them. Here we focus on the distinction that is made in Gallagher et al. (2019), between biases that are perceptual and biases that are non-perceptual in nature. As the reviewer points out, there is a possibly infinite number of bias manipulations that might produce either perceptual or non-perceptual effects. We cannot sensibly say anything about manipulations that we did not test in our experiment because we would not have data to substantiate those remarks. We merely show that base rate and payoff have similar effects which do not seem perceptual in nature, whereas the Müller-Lyer has a substantially different effect that does seem perceptual in nature. Further, we establish how these manipulations differentially affect confidence judgements. We now briefly point out in the discussion (line 679-683) that other bias types might produce different results and that it is difficult to generalize our results to untested bias manipulations.

- 7) Will reproductions always be possible?

Reproductions have always been a popular psychophysical task (they were not newly introduced to the field in 2023), both because they minimise bias and we are often interested in the perceptual experience of features that can easily be reproduced (e.g. tilt, or colour). However, we are also often interested in qualities (perceptual and cognitive) that might not be so easily reproduced in an experimental protocol.

What would the authors recommend as a means to gain insight into the underlying cause of a bias when a reproduction protocol is untenable?

Thank you for pointing this out. Indeed, we agree that reproduction tasks are not new, and we apologize if we have given the impression that we introduced them in 2023. Rather, they are a variant of a broader category of psychophysics tasks involved in measuring appearance as we now explain in the introduction (lines 106-108; for an excellent treatment of tasks that measure

appearance see Kingdom, Frederick. A. A. & Prins, N. (2016). *Psychophysics: A Practical Introduction* (Second Edition). Academic Press. <https://doi.org/10.1016/c2012-0-01278-1>). As the reviewer correctly points out, more complex perceptual and/or more abstract cognitive qualities may not be easily captured in a reproduction paradigm, or in other appearance paradigms such as those treated in the Kingdom book. Unfortunately, we do not have suggestions for better means to gain insight into these when a reproduction (or a different appearance protocol) is untenable. Indeed, we cannot even guarantee that reproduction is always the perfect assay for the difference between perceptual and non-perceptual effects even when this protocol is tenable. Investigating the mind is hard, we must row with the oars that we have and keep pushing for ways to improve.

Review comments

Reviewer #2 (Remarks to the Author)

Overall comments:

By integrating the standard perceptual decision-making paradigm with the controlled reproduction method, the manuscript effectively demonstrates how first-order decision and second-order confidence ratings are influenced by perceptual and non-perceptual manipulations. The research is grounded in robust methodology, and the findings hold significant value for the field. I only have relatively minor comments, which should be addressed to further strengthen the manuscript.

- 1) Visual inspection suggests that the point of maximal uncertainty (PMU) exhibited a larger shift than the point of subjective equality (PSE) under the perceptual manipulation, whereas the reverse appears true in the non-perceptual manipulations. These trends were consistent across Experiments 1 and 2, prompting me to consider the potential significance of this finding in the study of consciousness. If the authors deem it relevant, I would appreciate further discussion of this observation.

We want to start by thanking the reviewer for taking the time to read our manuscript and for the helpful comments that he or she provided. These comments have improved the manuscript considerably.

As Reviewer #2 correctly points out, while the Müller-Lyer illusion had a larger effect over the point of maximal uncertainty (PMU), the base rate and payoff manipulation had a larger effect on the point of subjective equality (PSE). As requested by Reviewer #1, we included a new analysis in which we employ ordinal modelling to directly test the relative effect of each manipulation on first-order decisions and confidence reports (Decision and confidence effects on page 14, line 477, note that line numbers refer to the revised document, not the tracked-changes document). We believe this new set of analysis should answer your concerns regarding the differential effects of the manipulations over first- and second-order decisions.

As suggested by the reviewer, we found an interaction in which perceptual and non-perceptual manipulations differentially load onto first-order and second-order decisions. Although confidence reports were influenced by non-perceptual information, the results suggest that when taken together with first-order responses, one could arbitrate between perceptual and non-perceptual manipulations. We performed this analysis on the combined data of experiments 1 and 2. However, the results were similar when we analysed the data of each experiment separately (see Supplementary Figure S14). We have also rewritten the discussion to discuss this result.

Note further that a small error in the initial analyses came to light when we redid our analyses. This error did not change any of the results, but we did update the manuscript to fix it, see for further explanation on the last page of this rebuttal.

- 2) The term "bias" is a bit overused, carrying different meanings depending on the context. For instance, "response bias" in line 49 exclusively refers to non-perceptual biases, such as those induced by base rate or payoff manipulations. Meanwhile, the "bias parameter" mentioned in line 51 pertains to the criterion in signal detection theory. Later, "biases in confidence" in line 101 describes shifts in empirical confidence data caused by whatever factors. Adopting consistent terminology would enhance readability, particularly for beginners.

Thank you for pointing this out. We modified the manuscript to use these terms consistently. We now use “bias” to refer to any kind of bias, regardless of the perceptual or non-perceptual nature of the manipulation, and “response bias” to refer to non-perceptual biases. We hope this will make the text more clear.

- 3) I was not quite following the paragraph starting from line 560. In my understanding, and as the authors noted in line 50, SDT serves only to dissociate sensitivity (d') from criterion (c) in decision-making. This only indicates these two parameters are estimated independently without carrying any further meaning. It is unfortunate that some researchers misinterpreted these parameters solely reflecting perceptual and non-perceptual effects. Yet, since SDT was never intended to distinguish between perceptual and non-perceptual effects, I find it even more difficult to grasp the intention behind this paragraph.

Indeed, as the reviewer correctly points out, SDT was originally devised to separate the ability of the participant to distinguish between signal and noise (d') from their overall tendency to prefer one of the options (criterion, or c). Also, as the reviewer points out, these two parameters are not intended to distinguish between the effects of a perceptual and a non-perceptual manipulation, as the criterion parameter can be shifted by both. This was pointed out in a paper by Witt et al. (2015). Although it has been some years since that paper was published, we believe it is good to mention, albeit briefly, that SDT was not meant nor can be used to distinguish between perceptual and non-perceptual effects. We have nevertheless rewritten the passage to make it more clear that SDT was never intended for such use.

Curve fitting quality on main results

While addressing the comments of reviewer #1 and #2 regarding the relative effect of the manipulations on first- and second-order decisions we realized we have missing values in some participants in the main analysis (Figure 2 and 3). The problem was that some participants were missing a point of subjective equality, a point of maximum uncertainty and/or a reproduction response equivalent reference line because their fits were not crossing the relevant point across the y-axis.

To address this issue, we have removed the data of participants with poor (insignificant) or inverted curve fits (opposite direction of the stimuli; e.g. 'short' responses increase as the target length increases). Participants missing one of these three data points (PSE, PMU or reference-reproduction) were removed from all the analyses. Removing the data of these participants did not change any of the main results. There is a decision and confidence effect of all manipulations but only the Müller-Lyer affects reproduction. The exact Bayes Factor and effect size values have been updated, as well as the participant count on figures 2 and 3. We have also described this procedure in the Methods – Participants and in the Analysis – Curve fitting sections.